# Tourism Development under Water-Energy Dual Constraints: A Case Study from Xinjiang Based on Different Emergency Scenarios

**DOI:** 10.3390/ijerph20032224

**Published:** 2023-01-26

**Authors:** Ruifang Wang, Fengping Wu, Zhaoli He

**Affiliations:** 1Business School, Hohai University, Nanjing 211100, China; 2School of Business Administration, Nanjing University of Finance and Economics, Nanjing 210046, China

**Keywords:** water pressure, energy consumption, tourism water footprint, tourism water supply, suitability, scenario analysis

## Abstract

The concept of green development requires that tourism development should be constrained by water and energy. This paper first constructed the calculation model of tourism water supply (TWS) based on water resources, economy, population, and employment. Second, according to the tourism life cycle theory, the energy-related water footprint account was built and combined with energy and water consumption, to realize water-energy dual constraints. Then, a suitability model between TWS and tourism water footprint (TWF) was established. Last, this paper predicted the growth rate of tourists in Xinjiang under the “suitability” state between TWS and TWF. Results show that in a future emergency-free setting, the average annual growth rate of tourists must be below 9.63% to maintain the “suitability” state, and in the context of emergencies damaging public health or socio-economic stability, the average annual growth rate may rise to 12.79%. In any scenario, the cap on tourist numbers in Xinjiang should be around 1.326 billion person-days in 2025, in line with the government’s planning goal. Last, this paper proposed suggestions to advance the green development of tourism from three angles: strengthening water conservation policies, promoting digital tourism, and setting multiple environmental monitoring mechanisms.

## 1. Introduction

Tourism, as an emerging pillar industry, is significant in promoting consumption, investment, employment, exchanges, and communication. Statista Research Department [1] suggests that the direct and total contribution of tourism to world GDP showed an obvious upward trend from 2006 to 2019, and surpassed 9 trillion dollars in 2019. In tourism output, China ranks the second in the world with 403.5 billion dollars, far ahead of other Asian and European countries [2].

However, current studies find that the intensive resources and the extensive development patterns of the tourism sector have inevitably led to excessive resource consumption and continuous environmental degradation, especially manifested in the more serious water stress, energy overuse, and higher pressure on carbon emission reduction [3,4,5].

Toth et al. [6] demonstrated that the gathering of tourists and certain water-consuming activities in both time and space will increase the pressure of local water supply. Tourism needs more fresh water not only for flushing toilets, replenishing swimming pools, and maintaining green space of hotels and attractions, but also for making high-grade food and producing fuel [7]. Many scholars even conclude that tourists consume two or three times as much water as local residents [8]. Especially in areas with water shortages and many tourists, their demand for water poses a bigger threat to sustainable development [4]. In a word, the sustainable development of tourist destinations mainly depends on the supply of water resources [9]. The large amount of greenhouse gas produced by fossil energy also puts great pressure on the green development of tourism [10,11]. Energy, mainly including raw coal, kerosene, and diesel [12], is used throughout tourism [13,14,15,16] for transportation, accommodation, and tourist activities [12]. Studies show that buses, private cars, airplanes, and other vehicles consume considerable energy [10,17,18], making transportation the largest energy consumer in tourism [19], and the transportation water footprint an integral part of the tourism water footprint (TWF) [20,21]. 

In addition, Yoon et al. [22] confirmed a strong correlation between water and non-renewable energy consumption in hotels and recreational activities in tourist resorts. In other words, energy virtual water is essential for establishing an integrated TWF measurement system. In the meantime, energy overuse also leads to higher greenhouse gas emission. Therefore, exploring the reasonable pace of tourism development from the angle of water-energy dual constraints helps further put the concepts of man-water harmony and low-carbon tourism into practice.

The number and growth rate of tourists are two critical indicators to demonstrate the status quo of tourism development. Although tourism is a pivotal engine for economic growth [23,24,25], the concept of sustainable development requires that the number of tourists to a tourist destination must be in line with the local resources and environment. In other words, once the number of tourists surpasses the threshold of the carrying capacity of the destination, it would cause ecological overloading and further affect the sustainable development of tourism and ecological civilization construction. Against the backdrop of global environmental crisis, it is ill-advised to only pursue a larger number of tourists. Instead, tourism development with environmental constraint should be the green development responsibility that every country shoulders, and the purpose for the international community to put the concepts of green development and man-water harmony into practice. As a result, water-energy dual constraints are a significant foundation to determine the reasonable growth rate of the tourist scale and promote regional ecological tourism, and a reference for administrative departments to take the “dual control actions” in managing water resources and energy.

This paper is structured as follows: part II summarizes the shortcomings of the existing relevant studies. In part III, tourism water supply (TWS) and TWF models, as well as a suitability model, are constructed to measure and demonstrate tourism water stress. Suitability is divided into six categories, among which “suitability” represents no water stress. Next, four scenarios of emergencies damaging public health or socio-economic stability are wet up with methods to predict TWS and TWF. After that, the growth rate of tourists is taken as an estimated value, and the corresponding calculation methods are proposed. Then, part III presents data resources and gives a brief introduction of Xinjiang. In part IV, this paper estimates the threshold of the average annual growth rate of tourists and TWF in Xinjiang in different scenarios during the 14th Five-Year Plan Period (2021–2025), with tourism water use in the “suitability” state. In part V, the results are compared with other studies to reveal deficiencies. Last, part Ⅵ concludes this paper and proposes some measures to promote the green development of tourism.

## 2. Literature Review

Scholars have explored how tourism development matches regional water resources and energy and had plenty of findings that fall into the following three categories: measurement of tourism water and energy demand, measurement of tourism water supply pressure, and adaptive management of tourist scale.

Life cycle assessment (LCA) is “a methodology to evaluate the environmental effects of a product or activity holistically, by analyzing the entire life-cycle of a particular product, process or activity” [26]. It has been extensively used in studies on tourism carbon and water footprint [21,27,28,29]. The footprint life cycle models based on LCA are also critical methods to measure tourism water and energy demand. To be specific, these models, according to the features of tourism life cycle, divide tourist activities into seven accounts of food, accommodation, transportation, visiting, shopping, entertainment, and waste disposal, and sum up each account to obtain the total tourism water or energy footprint. Gali et al. [30] proposed an index set called footprint family, including water footprint, energy footprint, carbon footprint, and others, serving as an objective assessment on human activities and revealing the resource consumption caused by production and life activities. On such a basis, some scholars point out that the tourism water footprint represents how much water is consumed during the production of products and services consumed in each stage of tourist activities in a certain area [31], while energy footprint represents how much energy is consumed during that process [12,13].

However, relevant studies create the water footprint accounts of six elements of tourism—food, accommodation, transportation, visiting, shopping, and entertainment [21,27], while few focus on a life cycle assessment of energy-related water footprint. Huang et al. [20] tried to include energy virtual water to measure the total TWF. He and Wang [31] created an energy-related water footprint account from the tourism life cycle theory, and estimated the account in Xinjiang according to the daily per capita consumption of energy. Moreover, other scholars measure the energy-related water footprint via direct and indirect water. For example, Li and Wang [32] and Li [33] both created an energy-related water footprint account containing energy consumed by accommodation and transportation. Meanwhile, the fuel water footprint calculated by Hadjikakou et al. [34] only involves energy consumed by airplanes, buses, and cars in the return trip.

Most of the current studies on tourism-related energy consumption focus on transportation, accommodation, and tourist activities [12,13,35]. Gössling [12] used a bottom-up approach to estimate the energy consumption in world tourism. The findings show that transportation consumes 13,223 PJ of energy (93.91% of the total consumption) while the accommodation consumes 508 PJ of energy (3.61%), and the rest, 350 PJ of energy, (2.48%) is consumed by activities. In 2011, Shi and Wu [13] first used a bottom-up approach to systematically estimate direct energy consumption and emission in Chinese tourism. The results show that the tourism in China has a direct energy consumption of 428.3 PJ in 2008 with 72.08% in tourism transportation and 22.60% in accommodation. Tao et al. [36] also used a bottom-up approach to estimate the tourism-related energy consumption in Jiangsu Province. It is shown that the energy used in transportation, accommodation, and tourist activities take up 70.91%, 17.32%, and 11.76%, respectively. 

In current research on tourism-related water stress, the ratio of the tourism water footprint (TWF) to tourism water supply (TWS) is taken as an index to measure water use intensity (WUI) or water resources carrying capacity (WRCC) [37,38]. However, a complete system to measure TWS stress has yet to be formed. Researchers Tian et al. [37] estimated TWS only according to the needs of residents and tourists for domestic water. Jie et al. [38] also made their calculations based on the historical data on the proportion of tourism water demand to total water supply.

In order to explore how the scale of tourists matches the resource endowment of the tourist destination, scholars have carried out extensive studies on the number of tourists that can be accommodated in a destination via tourism carrying capacity. The term can be defined as the maximum number of tourists that may visit a tourist destination at the same time without causing destruction of the physical, economic, and socio-cultural environment [39,40]. Relevant methods are primarily models of tourism carrying capacity calculation to obtain the cap of tourist number [41,42,43,44].

As mentioned above, in terms of matching tourist scale with resource endowment, there are still some deficiencies with the existing research. First, the cap of tourist number is mainly used to assess the potential of tourist destinations, while unable to intuitively reflect the speed of tourism development. Second, the research on tourism development speed from the perspective of water and energy dual constraints is scarce. In addition, as for the water stress of tourism, current studies are yet to establish a complete system for measuring tourism water supply. In addition, the current TWF account system only includes energy consumed by traffic. 

Compared with relevant researches, this paper makes the following contributions. First, it offers new insights for research on TWS. We constructed a calculation model of TWS in terms of local water resource endowment, tourism output, tourist number, and tourism practitioners. Second, it improves TWF account based on a life cycle assessment approach. Most studies ignore the energy-related virtual water (except transportation fuel consumption) in tourism. However, we incorporated all the energy virtual water by constructing an energy-related water footprint account to make it more accurate to measure TWF. Finally, this paper enriched research methods on the green development of tourism via water-energy dual constraints. We established a suitability model to measure the tourism water stress, and integrated energy-related water footprint into TWF, which indirectly curbs energy consumption when water stress is in “suitability” state.

## 3. Materials and Methods

This paper aims to achieve a reasonable range of the growth rate of tourists with constraint on local resources. It refers to the average annual growth rate of tourists in years without emergencies affecting tourism development. Detailed steps are as follows:**Step 1:** Constructing calculation models on TWS and TWF. First, this paper constructed a TWS model based on local water resource endowment, economic development, tourist number, and employment. Second, TWF was applied to the measurement of tourism water demand, with the tourism life cycle theory to create water footprint accounts related to food, accommodation, energy, visiting, and shopping.**Step 2**: Building a suitability model between TWS and TWF to measure tourism water stress. The stress was quantified by water demand and supply, then divided into six catalogues including the “suitability” state.**Step 3**: Setting up scenarios. Four scenarios with emergencies affecting the tourism industry were set up to predict the prospect of tourism and the future growth rate of tourists based on the average annual growth rate of tourists in each scenario.**Step 4**: Predicting the growth rate of tourists without emergencies. This paper predicts the development indicators related to tourism in the study area in the future, then figures out the growth rate of tourists in different scenarios in accordance with the value of TWF/TWS under the state of “suitability”.

### 3.1. Calculation Models of TWS and TWF

In the light of the definition of tourism water resources by Jie et al. [38] and He and Wang [31], the suitability between TWS and TWF in this paper is defined as how much the water that local tourism needs matches the water that can be allocated to tourism in total water supply, under certain levels of scientific development. It is also the quantification of tourism water pressure.

#### 3.1.1. Calculation Model of TWS

At present, there are not enough studies on measuring water supply for tourism or scenic spots. In line with Jie et al. [38] and Wang et al. [45], how much water allocated to tourist attractions from the local water supply is mainly affected by local resource endowment and tourism economy. Therefore, this paper used a causal model to predict TWS, taking into account the proportion of tourists in the local population, the contribution rate of tourism to the local GDP, and the proportion of tourism practitioners in local employed population. The causal model (1) is as follows:(1)$$TWS=(w_{1}\times N_{v}D_{v}/Pop+w_{2}\times g/G+w_{3}\times n_{emp}/N_{emp})\times WS_{total}$$
where w is the weight; $N_{v}$ and $D_{v}$ represent the number of tourists (unit: person) and length of stay (unit: day), so $N_{v}D_{v}$ is the total tourist number (unit: person-day); Pop is resident population (unit: person-day); g is tourism consumption; G is local GDP; $n_{emp}$ represents tourism practitioners, including staff of star-rated hotels and travel agencies, and full-time staff of A-grade tourist attractions; $N_{emp}$ represents employed population; and $WS_{total}$ is the local annual water supply—local water resource endowment.

#### 3.1.2. Calculation Model of TWF

Based on the theory of footprint family [30], TWF in this research indicates the tourism water demand. In line with the TWF model put forth by Huang et al. [20] and He and Wang [31], five water footprint accounts: food, accommodation, energy, visiting, and shopping were taken into account in this paper. Moreover, energy is used throughout the tourism life cycle, and transportation water footprint is composed of the energy virtual water of various transportation modes. Hence, this paper integrated transportation water footprint into the energy-related account. TWF is represented as follows:(2)$$TWF=WF_{food}+WF_{accom}+WF_{ener}+WF_{visit}+WF_{shop}$$
where TWF is the total TWF; $WF_{food}$, $WF_{accom}$, $WF_{ener}$, $WF_{visit}$, and $WF_{shop}$ represent the TWFs with regard to food, accommodation, energy, visiting, and shopping. Among them, the energy-related account includes all the virtual water for raw coal, kerosene, and diesel during the tourism life cycle.

In tourism, the energy-related water footprint is mainly caused by transportation, accommodation, and catering [13,19]. Meanwhile, modes of modern transportation are mostly railway, highway, civil aviation and water way. The unit energy consumption of each mode and the proportion of tourists in turnover volume of passenger traffic are shown in Table 1.

According to Table 1, the energy consumed by tourism transportation is: (3)$$E_{tran}=\sum_{i=1}^{3}\gamma_{i}T_{i}\eta_{i}$$
where $E_{tran}$ (unit: MJ) is the total energy consumed by transportation; i = 1, 2, 3, and 4 represent the four kinds of transportation modes—railway, highway, civil aviation, and water way; γ is the proportion of tourists in turnover volume of passenger traffic; T is passenger turnover volume (passenger-kilometer); η is energy consumption intensity (MJ/passenger-kilometer).

In this paper, unit energy consumption of a three-star hotel—110 MJ/bed/night [19] was taken as the average energy consumption per night of overnight tourists. Thus, the accommodation energy consumed by overnight tourists is estimated as:(4)$$E_{\mathrm{accom}}=110\times {\;Q}_{v}D_{v}$$
where $E_{tran}$ (unit: MJ) is the total energy consumed by overnight tourists for air conditioning, washing, watching TV, etc.; $Q_{v}$ and $D_{v}$ are the number of overnight tourists (unit: person) and the length of stay (unit: night). Therefore, $Q_{v}D_{v}$ is the total number of overnight tourists (unit: person-night).

In line with the research of Yang et al. [47], this paper took 48.7 MJ/person/time as the energy consumption intensity of hotel catering service. So, the energy consumed by catering is as follows:(5)$$E_{cater}=48.7n_{food}N_{v}D_{v}$$
where $E_{cater}$ (unit: MJ) is the total energy needed to cook for tourists; $n_{food}$ indicates the number of meals; $N_{v}D_{v}$ represents the total number of tourists (see Formula (1)).

Next, the energy consumption is converted into standard coal (1PJ = 34,129.69 tons of standard coal, according to the United Nations Convention), then into raw petroleum (1 t raw petroleum = 1.4286 tons of standard coal), and finally into an energy-related water footprint (1.06 m³ water /1 GJ raw petroleum) (unit: 100 million m³):(6)$$WF_{ener}=(\sum_{i=1}^{3}\gamma_{i}T_{i}\eta_{i}+110Q_{v}D_{v}+48.7n_{food}N_{v}D_{v})\times 34129.69\times 1.06\times 10^{-6}/1.4286$$

The remaining four water footprint accounts are be calculated as shown in Table 2.

### 3.2. Suitability Model between TWS and TWF

With y=TWF/TWS, f(y) as the indicator to measure the suitability between TWS and TWF, this paper applied the exponential function [48] to the calculation model of suitability. The model is as follows:(7)$$f(y)=\left\{\begin{array}{cc} c_{1}\times (1-e^{1-y}), & a\leq y\leq 1\\ c_{2}\times (1-e^{y-1}), & 1<y\leq b \end{array}\right.$$
where $y\in \left[a,b\right]$, c>0, a>0, f(a)=−100, and f(b)=100. When the suitability value f(y)=0, water supply equals demand, indicating no water pressure. When f(y)<0, water demand is less than supply, indicating a great potential for tourism development with enough water supply. When f(y)>0, water demand exceeds supply, indicating potential water pressure. Details are shown in Figure 1.

Suitability is divided into six categories, as shown in Formula (8): (8)$$\left\{\begin{array}{cccccccc} -100 & < & f(y) & \leq & \delta_{1} & , & A & \\ \delta_{1} & < & f(y) & \leq & \delta_{2} & , & B & \\ \delta_{2} & < & f(y) & \leq & \delta_{3} & , & C & \\ \delta_{3} & < & f(y) & \leq & \delta_{4} & , & D & \\ \delta_{4} & < & f(y) & \leq & \delta_{5} & , & E & \\ \delta_{5} & < & f(y) & \leq & 100 & , & F & \end{array}\right.$$
where $\delta_{1},\delta_{2}<0$ and $\delta_{3},\delta_{4},\delta_{5}>0$. In this paper, the value of δ is determined by expert advice, local tourism water consumption over the years, and local government’s policy on tourism. The detailed categories are shown in Table 3.

### 3.3. Setting Up Scenarios

Emergencies, such as pandemic disease, financial crisis, and war, would seriously affect the number of tourists and tourism revenue [49,50,51]. In this paper, four scenarios were set up according to whether there are emergencies damaging public health, or socio-economic stability between 2019 and 2025, as shown in Table 4.

In Scenarios II, III, and IV, the probability of normal development was set as *P*_1_, as shown in Formula (9). In normal development years, the growth rate of tourists $\alpha_{1i}$ (*I* = 1, 2, 3, and 4 represent four scenarios) changes with the scenarios and is to be predicted. Likewise, the probabilities of emergencies in Scenarios II, III, and IV were set as *P*_2_, *P*_3_, and *P*_4_, as shown in Formula (9). This study aimed to obtain the average annual growth rate of tourists under normal development ($\alpha_{1i}$), so the growth rate with emergencies in Scenario II was set as the mean value $\alpha_{2}$, as shown in Formula (10). This formula can also be applied to Scenarios III and IV.
(9)$$P_{i}=n_{year}^{\left(i\right)}/N_{year},i=1,2,3,4$$
(10)$$\alpha_{i}=\sum_{j=1}^{N_{year}}r_{ij}/m,i=2,3,4$$

In Formula (9), $n_{year}^{\left(i\right)}$ represents the number of years in which event i occurs; $N_{year}$ represents the number of sample years. In Formula (10), $r_{ij}$ represents the growth rate of tourists when emergency i occurs in year j; *m* represents, during the sample period, the number of years in which emergency i occurs.

In Scenario IV, if the two types of emergencies do not happen in the same year, *P*_4_ and $\alpha_{4}$ can be estimated as follows:(11)$$\begin{array}{cc} P_{4}=P_{2}\times P_{3}, & \alpha_{4}=\min\left\{\alpha_{2},\alpha_{3},\left(\alpha_{2}+\alpha_{3}\right)/2\right\} \end{array}$$

### 3.4. Methods to Predict the Growth Rate of Tourists in Normal Development Years

#### 3.4.1. Predicting Indicators Related to Water Supply and Demand

This paper regards the growth rate of overnight tourists as the same as that of all tourists. According to Formulas (9) and (10), this paper estimated the total number of tourists ($N_{v}D_{v}$) and overnight tourists ($Q_{v}D_{v}$) in the forecast year, as shown in Formulas (12) and (13):(12)$$N_{v}D_{v}=\left\{\begin{array}{ll} {\left(1+\alpha_{11}\right)}^{n}N_{v}^{\left(0\right)}D_{v}^{\left(0\right)} & ,scenario\;I\\ P_{1}{\left(1+\alpha_{12}\right)}^{n}N_{v}^{\left(0\right)}D_{v}^{\left(0\right)}+P_{2}{\left(1+\alpha_{2}\right)}^{n}N_{v}^{\left(0\right)}D_{v}^{\left(0\right)} & ,scenario\;\mathrm{II}\\ P_{1}{\left(1+\alpha_{13}\right)}^{n}N_{v}^{\left(0\right)}D_{v}^{\left(0\right)}+P_{3}{\left(1+\alpha_{3}\right)}^{n}N_{v}^{\left(0\right)}D_{v}^{\left(0\right)} & ,scenario\;\mathrm{III}\\ P_{1}{\left(1+\alpha_{14}\right)}^{n}N_{v}^{\left(0\right)}D_{v}^{\left(0\right)}+P_{4}{\left(1+\alpha_{4}\right)}^{n}N_{v}^{\left(0\right)}D_{v}^{\left(0\right)} & ,scenario\;\mathrm{IV} \end{array}\right.$$(13)$$Q_{v}D_{v}=\left\{\begin{array}{ll} {\left(1+\alpha_{11}\right)}^{n}Q_{v}^{\left(0\right)}D_{v}^{\left(0\right)} & ,scenario\;I\\ P_{1}{\left(1+\alpha_{12}\right)}^{n}Q_{v}^{\left(0\right)}D_{v}^{\left(0\right)}+P_{2}{\left(1+\alpha_{2}\right)}^{n}Q_{v}^{\left(0\right)}D_{v}^{\left(0\right)} & ,scenario\;\mathrm{II}\\ P_{1}{\left(1+\alpha_{13}\right)}^{n}Q_{v}^{\left(0\right)}D_{v}^{\left(0\right)}+P_{3}{\left(1+\alpha_{3}\right)}^{n}Q_{v}^{\left(0\right)}D_{v}^{\left(0\right)} & ,scenario\;\mathrm{III}\\ P_{1}{\left(1+\alpha_{14}\right)}^{n}Q_{v}^{\left(0\right)}D_{v}^{\left(0\right)}+P_{4}{\left(1+\alpha_{4}\right)}^{n}Q_{v}^{\left(0\right)}D_{v}^{\left(0\right)} & ,scenario\;\mathrm{IV} \end{array}\right.$$where $N_{v}^{\left(0\right)}D_{v}^{\left(0\right)}$ and $Q_{v}^{\left(0\right)}D_{v}^{\left(0\right)}$ represent the total number of tourists and overnight tourists in the base year, respectively. 

Assume:(14)$$f(\alpha )=\left\{\begin{array}{ll} {\left(1+\alpha_{11}\right)}^{n} & ,scenario\;I\\ P_{1}{\left(1+\alpha_{12}\right)}^{n}+P_{2}{\left(1+\alpha_{2}\right)}^{n} & ,scenario\;\mathrm{II}\\ P_{1}{\left(1+\alpha_{13}\right)}^{n}+P_{3}{\left(1+\alpha_{3}\right)}^{n} & ,scenario\;\mathrm{III}\\ P_{1}{\left(1+\alpha_{14}\right)}^{n}+P_{4}{\left(1+\alpha_{4}\right)}^{n} & ,scenario\;\mathrm{IV} \end{array}\right.$$

Then:(15)$$\left\{\begin{array}{l} N_{v}D_{v}=N_{v}^{\left(0\right)}D_{v}^{\left(0\right)}\times f(\alpha )\\ Q_{v}D_{v}=Q_{v}^{\left(0\right)}D_{v}^{\left(0\right)}\times f(\alpha ) \end{array}\right.$$

Methods to predict other relevant indicators are shown in Table 5.

According to Table 5 and Formulas (1) and (15), TWS can be estimated as Formula (16):(16)$$TWS=w_{1}\frac{N_{v}^{\left(0\right)}D_{v}^{\left(0\right)}f(\alpha )}{k/\left(1+be^{\left(-at\right)}\right)}+w_{2}\frac{c_{0}+c_{1}t+c_{2}t^{2}}{{\left(1+\alpha_{G}\right)}^{t}G_{0}}+w_{3}\frac{n_{emp}^{\left(0\right)}+t\Delta n_{emp}}{{\left(1+\alpha_{Nemp}\right)}^{t}N_{emp}^{\left(0\right)}}$$

Based on Table 2 and Table 5, Formulas (6) and (15), each TWF account in the forecast year can be calculated as follows:(17)$$WF_{food}=[N_{v}^{\left(0\right)}D_{v}^{\left(0\right)}f(\alpha )+D_{emp}(n_{emp}^{\left(0\right)}+t\Delta n_{emp})]\times (\sum_{j=1}^{n}\sum_{k=k_{0}}^{k_{0}+t-1}food_{jk_{0}}w_{j}^{food}/t\times +0.0015)$$
(18)$$WF_{accom}=q_{v}Q_{v}^{\left(0\right)}D_{v}^{\left(0\right)}f(\alpha )+q_{emp}D_{\mathrm{emp}}^{accom}n_{emp}^{accom}+\sum_{k=1}^{n}P_{k}M_{k}$$
(19)$$WF_{energy}=0.0253[\sum_{i=1}^{3}\gamma_{i}\eta_{i}(1+\alpha_{i}^{T})T_{i}^{\left(0\right)}+110Q_{v}^{\left(0\right)}D_{v}^{\left(0\right)}f(\alpha )+48.7n_{food}N_{v}^{\left(0\right)}D_{v}^{\left(0\right)}f(\alpha )]$$
(20)$$WF_{shop}=w_{indu}(c_{0}+c_{1}N_{v}^{\left(0\right)}D_{v}^{\left(0\right)}f(\alpha )+c_{2}Inc)$$

In addition, an increasingly large number of tourists require more water for public toilets. Therefore, according to He and Wang [31], this paper estimated the water for future tourist activities based on the increase of water for public toilets. Here is the formula of the visiting water footprint:(21)$$WF_{visit}=N_{v}^{\left(0\right)}D_{v}^{\left(0\right)}(f(\alpha )-1)w_{p}+WF_{visit}^{\left(0\right)}$$
where $w_{p}$ is water supply for public toilets; $WF_{visit}^{\left(0\right)}$ represents the visiting water footprint in the base year.

From Equations (17)–(21), we can see that all water footprint accounts are functions of f(α). Therefore, TWF can be described as TWF(f(α)).

#### 3.4.2. Steps to Predict the Growth Rate of Tourists

Water used for tourism in the future should always be kept in “suitability” state. In different scenarios, the growth rate of tourists in normal development years ($\alpha_{1i}$) is different. Suppose $\alpha_{2},\alpha_{3}$ and $\alpha_{4}$ are given, f(α) can be represented as $f(\alpha_{1i})$. Then, TWS and TWF can be described as functions of $\alpha_{1i}$. Therefore, $y=TWF(f(\alpha_{1i}))/TWS(f(\alpha_{1i}))=y(\alpha_{1i})$ is obtained. The steps to predict $\alpha_{1i}$ are as follows:**Step 1**: Calculating the TWS and TWF in the forecast year. Tourism water supply in the study area in four scenarios can be predicted according to Table 5 and Formula (16), and TWF according to Table 2 and Table 5, and Formulas (2) and (17)–(21).**Step 2**: Determining the ratio of water demand to supply in “suitability” state. In “suitability” state, the range of f(y) is $\left[\delta_{2},\delta_{3}\right]$ (see Formula 8 and Table 3). By solving the inequality $\delta_{2}\leq f(y)$, $\underline{y}$ (the lower limit of the ratio of water demand to supply y) is obtained. Likewise, $\bar{y}$ (the upper limit of y) is figured out with the inequality $f(y)\leq \delta_{3}$.**Step 3**: Making clear of the range of the growth rate of tourists in normal development years in each scenario. Based on Formulas (2) and (16)–(21) and $\underline{y}$, $\underline{\alpha_{1i}}$ (the lower limit of the average annual growth rate of tourists in normal development years) can be figured out. In the same way, $\bar{\alpha_{1i}}$ (the upper limit) can also be worked out based on $\bar{y}$. In addition, i=1,2,3,4 represents four scenarios. 

### 3.5. Study Area and Data Source

#### 3.5.1. Overview of the Study Area

Xinjiang, as a typical arid and semi-arid region in China, enjoys abundant tourism resources yet suffers a contradiction between rapid tourism development and environmental vulnerability. According to the *Statistical Yearbook of Chinese Cultural Relics and Tourism (2020)*, in 2019, Xinjiang had 496 A-rated scenic spots (including 13 AAAAA-level ones), and 12 ancient cities listed as World Cultural Heritages. The number of A-rated scenic spots in Xinjiang ranks in the top 10 in China. In recent years, the Chinese government has formulated a series of strategic plans and policies to facilitate tourism in Xinjiang. According to the Ministry of Culture and Tourism of China (2016–2020) and National Tourism Administration of China (2001–2018), from 2001 to 2019, the year-on-year growth rate of domestic tourists in Xinjiang went up, even reaching 40.77% in 2019. Additionally, the contribution of tourism consumption to Xinjiang’s GDP also increased from 5.36% in 2001 to 26.73% in 2019. However, according to *China Water Resources Bulletin*, the rainfall was only 174.7 mm in 2019, ranking low across China. In the meantime, according to *China Statistical Yearbook on Environment*, Xinjiang had only 4.87% of forest coverage rate in 2019, the only province below 5%, whereas it had 75 million hectares of sandy land, ranking the highest in China. It is thus clear that the rapid development of tourism in Xinjiang is in direct contradiction with its environmental vulnerability. Tourism in Xinjiang is in desperate need of achieving sustainable development at a reasonable speed. 

In terms of tourism elements, accommodation, transportation, visiting, and entertainment mostly consume virtual water and physical water within the region. As for “catering” for tourists, beef, mutton, milk, and eggs are basically produced in the region, with the exceptions of some soybeans and pork. As for “shopping”, commodities—mostly local specialties, handmade souvenirs, and jade—are also made locally, except for a few goods purchased. Therefore, this paper assumed tourism-related water is only consumed within Xinjiang.

#### 3.5.2. Data Source

The sample period is 2001–2019, with 2019 as the base year for prediction. The growth of tourists in Xinjiang during the sample period is shown in Figure 2. Strategy of Revitalizing Xinjiang through Tourism is a major part of the 14th Five-Year Plan (2021–2025) for socioeconomic development of Xinjiang. Therefore, this paper took the year 2025 (the end of the 14th Five-Year Plan Period) as the forecast year.

Basic data is composed of two parts—official statistics and quotas:Statistics are from the Ministry of Culture and Tourism of China (2016–2020), National Tourism Administration of China (2001–2018), Statistic Bureau of Xinjiang Uygur Autonomous Region (2001–2020), Department of Water Resources of Xinjiang Uygur Autonomous Region (2001–2018), and Ministry of Water Resources of China (2002–2020).Quotas are from GB50015-2003 Code for Design of Building Water Supply and Drainage, and Industrial and Domestic Water Quota of Xinjiang Uygur Autonomous Region.

## 4. Results

### 4.1. Prediction Results of TWS and TWF

#### 4.1.1. TWS at the End of the 14th Five-Year Plan Period

The sample period (2001–2019) contains 16 years of normal development years. Thus, we set the probability 16/19 as $P_{1}$, as seen in Formula (9). The SARS outbreak of 2003 menaced the world public health system, and the tourist number of Xinjiang only had a year-on-year increase of 4.95%. In addition, the global financial crisis in 2008 and 2009 undermined the socioeconomic stability of Xinjiang, bringing the growth rate of tourists down to −1.54% and −30.08%, respectively. Therefore, according to Formulas (9) and (10), the probability of emergencies damaging public health was set as 1/19 ($P_{2}$), with the growth rate of tourists as 4.95% ($\alpha_{2}$); the probability of emergencies damaging socioeconomic stability was set as 2/19 ($P_{3}$), with the growth rate of tourists as (−1.54% − 30.08%)/2=−15.81% ($\alpha_{3}$). However, two types of such emergencies did not occur in the same year during the sample period. Thus, based on Formula (11), the future probability of these two types of emergencies happening in the same year was 1/19×2/19 ($P_{4}$), with the growth rate of tourists $\min\left\{4.95\%,-15.81\%,(4.95\%-15.81\%)/2\right\}$ ($\alpha_{4}$). In conclusion: (22)$$f(\alpha )=\left\{\begin{array}{ll} {\left(1+\alpha_{11}\right)}^{6} & ,scenario\;I\\ 16/19\times {\left(1+\alpha_{12}\right)}^{6}+1/19\times {\left(1+4.95\%\right)}^{6} & ,scenario\;\mathrm{II}\\ 16/19\times {\left(1+\alpha_{13}\right)}^{6}+2/19\times {\left(1-15.81\%\right)}^{6} & ,scenario\;\mathrm{III}\\ 16/19\times {\left(1+\alpha_{14}\right)}^{6}+1/19\times 2/19\times {\left(1-15.81\%\right)}^{6} & ,scenario\;\mathrm{IV} \end{array}\right.$$

Based on Formula (15), the total number of tourists and overnight tourists (unit: 100 million person-days) in Xinjiang in 2025 can be formulated as
(23)$$\left\{\begin{array}{l} N_{v}D_{v}=7.64f(\alpha )\\ Q_{v}D_{v}=7.63f(\alpha ) \end{array}\right.$$

In the light of the predicting methods in Table 5, the predicted values of other water supply indicators are shown in Table 6.

According to expert advice and actual water supply in Xinjiang, $w_{1},w_{2}$ and $w_{3}$ in Formula (1) are assigned 0.17, 0.79, and 0.04. And according to Table 6, the predicted results of TWS in 2025 in different scenarios are shown in Formula (24):(24)TWS=6.7607f(α)+99.1099

#### 4.1.2. TWF at the End of the 14th Five-Year Plan Period

According to Formulas (2)–(6) and Table 2, Figure 3a is drawn and shows the TWF of Xinjiang during the sample period. In general, water demand rises up steadily. In 2019, it reaches 9.4 billion m³, among which the water footprints of food, accommodation, energy, visiting, and shopping are 1.093, 0.267, 5.974, 1.310, and 0.756 billion m³, respectively. In addition, the energy-related water footprint always makes up the largest proportion of the total TWF (see Figure 3b), even reaching 82.56% in 2014. After that, it drops down, yet still remaining above 50%. The proportion of catering energy water footprint gradually increases, making it the largest energy consumer. While the overall energy consumption of transportation is increasing steadily, its contribution to the energy and water footprint is decreasing year by year.

The predicted values of the relevant indicators of TWF in 2025 are shown in Table 7 (based on calculation methods in Table 2 and Table 5). According to Formulas (2) and (17)–(21), TWF in 2025 is calculated as follows:(25)TWF=72.7760f(α)+24.1445

### 4.2. Determining the Parameters in the Suitability Model

Tourism water resource carrying capacity (WRCC) usually faces severe overload when TWF reaches twice of water supply [38]. Therefore, this paper set the value of *b* in Formula (7) as two. Next, because the ratio of water consumption to available water resources in most arid areas of northwest China is between 0.12 and 0.9 [53], this paper set the value of *a* in Formula (7) as 0.12. Setting f(0.12)=−100,f(2)=100,f(1)=0, Formula (7) was solved accordingly to obtain $c_{1}=70.8768,c_{2}=-58.1977$. As a result, the suitability model for *TWF*/*TWS* in Xinjiang is as follows:(26)$$f(y)=\left\{\begin{array}{cc} 70.8768\times (1-e^{1-y}), & 0.12\leq y\leq 1\\ -58.1977\times (1-e^{y-1}), & 1<y\leq 2 \end{array}\right.$$

Again, according to expert advice and the status quo of tourism industry in Xinjiang, the value ranges of f(y) and y in different suitability categories are determined (see Table 8), with y in “suitability” state as $\left[0.6979,1.3574\right]$.

### 4.3. Growth Rate of Tourist Number in 2025

In line with Formulas (24) and (25), in “suitability” state, the value range of the ratio of water demand to supply—$\underline{y}\leq y\leq \bar{y}$ is expressed as follows:(27)$$\left\{\begin{array}{l} \begin{array}{c} y=\frac{72.7760f(\alpha )+24.1445}{6.7607f(\alpha )+99.1099}\\ 0.6979\leq y\leq 1.3574 \end{array}\\ f(\alpha )=\left\{\begin{array}{ll} {\left(1+\alpha_{11}\right)}^{6} & ,scenario\;I\\ 16/19\times {\left(1+\alpha_{12}\right)}^{6}+1/19\times {\left(1+4.95\%\right)}^{6} & ,scenario\;\mathrm{II}\\ 16/19\times {\left(1+\alpha_{13}\right)}^{6}+2/19\times {\left(1-15.81\%\right)}^{6} & ,scenario\;\mathrm{III}\\ 16/19\times {\left(1+\alpha_{14}\right)}^{6}+1/19\times 2/19\times {\left(1-15.81\%\right)}^{6} & ,scenario\;\mathrm{IV} \end{array}\right. \end{array}\right.$$

Formula (27) indicates that to keep tourism water in a “suitability” state in all four scenarios, the range of growth rate of tourists in normal development years in the future should be maintained at:(28)$$\left\{\begin{array}{ll} -6.65\%\leq \alpha_{11}\leq 9.63\% & ,\;\mathrm{scenario}\;I\\ -5.72\%\leq \alpha_{12}\leq 12.04\% & ,\;\mathrm{scenario}\;\mathrm{II}\\ -4.87\%\leq \alpha_{13}\leq 12.40\% & ,\;\mathrm{scenario}\;\mathrm{III}\\ -3.99\%\leq \alpha_{14}\leq 12.79\% & ,\;\mathrm{scenario}\;\mathrm{IV} \end{array}\right.$$

By putting Formula (28) into Formulas (22)–(25), tourist number, TWS, and TWF in 2025 in a “suitability” state can be obtained (see Table 9).

## 5. Discussion

### 5.1. Results Discussion

#### 5.1.1. Scenario Analysis

In the case of no emergencies, the average annual growth rate of tourists must be maintained below 9.63% during the 14th Five-Year Plan Period to ensure the sustained development of economy and water resources. If the growth rate exceeds 9.63%, the ratio of TWF to TWS will surpass 1.3574, resulting in “overload” in tourism water use. However, due to the uncertainty of emergencies, the upper limit of tourist growth rate may vary before and after the emergencies. In Scenario Ⅱ, the growth rate of tourists must be kept below 9.63% before emergencies, to keep tourism water in a “suitability” state. However, it can be elevated to 12.04% after the emergencies. Similarly, in Scenarios III and Ⅳ, the growth rate of tourists after emergencies can be raised to 12.40% and 12.79%, respectively. Furthermore, the upper limit of the tourist growth rate in each scenario is basically consistent with the findings of Wu et al. [52].

The number of tourists may show negative growth. In Scenario I, tourism water stress will always be in a “suitability” state as long as the average annual growth of tourists is higher than −6.65%. However, negative growth is unlikely to happen in Scenario I because, without emergencies affecting tourism development, local economy would drive the tourism industry. However, once emergencies occur, tourist number may show negative growth. Thus, in Scenarios II, III, and IV, to keep tourism water pressure in a “suitability” state, the growth rate of tourists must remain above −5.72%, −4.87%, and −3.99%, respectively. Then again, if tourist number drops dramatically and its growth rate breaks through the lower limit, the tourism industry will be “underdeveloped”. This means that the local tourism industry under current water supply capacity still has a long way to go for development. 

No matter in which scenario, the number of tourists in Xinjiang in 2025 must be kept at 0.505–1.326 billion person-days, to keep TWF and TWS in a “suitability” state. The Chinese government has set a goal of receiving 400 million tourists at home and abroad (1.38 billion person-days, based on the average stay of 3.45 days) by the end of the 14th Five-Year Plan Period. This is very close to the upper limit in this paper, 1.326 billion person-days, proving that the prediction methods in this paper has certain applicability.

#### 5.1.2. Analysis of Water Footprint Account

During the sample period, water use for tourism in Xinjiang exceeded 10 m^3^ per capita per day. Apart from practitioners, each tourist consumed 12.27 m³ every day in 2019, much higher than 2–7.5 m³ proposed by Gössling et al. [7]. The reason is that Xinjiang’s per capita water consumption is about five times that of the national level, due to its high water consumption rate for social and economic development. As a result, per tourist water consumption in Xinjiang is also much higher than that of other areas of China and even other countries, but the number in 2019 (12.27 m³/day) is only twice 6.43 m³/day—local per resident water consumption. To be specific, the food water footprint is 1.43 m³ per tourist per day, slightly higher than that in Mount Huangshan (1.17 m³) [27], mainly due to different eating habits. For example, tourists in Mount Huangshan prefer pork, while visitors in Xinjiang prefer beef and mutton, which contain more virtual water. Per capita water footprint of accommodation in Xinjiang is 350.66 L/day, much higher than 292 L/day of Penghu Island [19], mainly because of different water quota for hotels. Meanwhile, in Xinjiang, three-star hotels can use 350 L of water for each person every night.

This paper found that energy-related water footprint is a major part of TWF, consistent with the findings of Li and Wang [32] and Li [33]. However, what is totally opposite is that this paper found that energy-related water footprint is larger than the food water footprint. The reason is that in this paper, energy-related water footprint account includes virtual water used for transportation, catering, and accommodation, leading to more comprehensive results. In addition, a yearly increasing energy-related water footprint more or less represents the upward trend of energy consumption. Statistics show that energy-related water footprint of tourism in Xinjiang reaches 5.974 billion m^3^ in 2019, with 308.86 MJ/tourist/day of per capita energy consumption—much lower than the 484 MJ/tourist/day of Penghu Island in summer [19]. There are two reasons. First, study of Kuo and Chen [19] show that 97.5% tourists go to Penghu Island by plane (high water-consumption intensity), so the plane takes the lion’s share of energy consumed by transportation. However, in Xinjiang, the plane only takes up 65% of the transportation energy. Second, Kuo and Chen [19] measure energy used for transportation, accommodation, and recreational activities such as visiting historical sites and scenic spots. However, this paper only estimates energy consumption in transportation, accommodation, and catering, leading to relatively different results.

### 5.2. Limitations and Future Research Directions

As tourism is a complex system that integrates social economy and ecology, it is complicated to accurately estimate the speed of tourism development. Most studies evaluate tourism development under one or several constraints of water resources carrying capacity, efficiency of energy use, land use pattern, waste treatment ability, and so on [54,55,56,57]. Similarly to their studies, this paper only makes a tentative exploration into the speed of tourism development under water-energy dual constraints. We will take multiple constraints into thorough consideration from the angle of social economy and ecology, including waste treatment ability, landscape attraction, resource endowment, and so on in the future.

Other tourism activities, such as sightseeing, vacation, and visiting relatives, as well as waste disposal, also consume water and energy [19], but they are not estimated in this paper. This is also a problem that many other TWF studies ignored [21,27], although it is a necessary part of TWF. Consequently, the measured TWF is lower than the actual value. The question of how to measure water and carbon footprint of tourism in a more comprehensive way is fundamental for virtual water strategy and future research.

The probability of emergencies, a factor influenced by sample period, medical level, national security, and ecological damage, affects the tourist number and total tourism consumption. Due to insufficient statistical data, this paper took the proportion of emergency years in the sample years as the emergency probability, to predict the impact of emergencies on tourist number. With the improvement of the data, mathematical models should be constructed to accurately estimate the possibility of sudden events and their impact on the scale of tourists.

## 6. Conclusions

To make big advances in green development and human-water harmony, this paper predicted the reasonable growth rate of tourists in Xinjiang in 2025. For that, we constructed TWS and TWF models, and a suitability model, and found that at the end of the 14th Five-Year Plan Period, Xinjiang’s tourist number cap, in whichever scenario, should be at 0.505–1.326 billion person-days. If the number is below 0.505 billion person-days, the tourism sector would be “underdeveloped”, indicating that TWS can support more tourists. On the other hand, if the number exceeds 1.326 billion person-days, TWS will be “overloaded”, and tourism water supply cannot support more tourists. However, the growth rate of tourists varies in each scenario. To be specific, without emergencies, the rate must be capped at about 9.63%; in case of emergencies damaging public health or socioeconomic stability, it can be elevated to about 12.79%.

The growth rate of tourists in this paper is grounded in the current resource consumption level. Driven by green development, low-carbon tourism, and sustainable development, this paper, in order to further promote tourism in Xinjiang and reduce the intensity of tourism resource consumption, proposes suggestions in the following three categories: strengthening water conservation policies, promoting digital economy, and setting multiple environmental monitoring mechanisms:Strengthening water conservation policies to intensify efforts to save water. First, implementing the Strictest Water Resources Management System (SWRMS) to adjust water supply. To do that, the government could encourage tourism enterprises to use recycled water for green belts conservation and road cleaning. For those tourism enterprises that exceed the planned water use, the relevant government departments should directly restrict their water use and force the person in charge to take corresponding responsibility. Second, improving the water-saving facilities to increase the annual average water conservation rate. The quota water consumption for guest rooms is vigorously encouraged to be implemented in hotel. Besides, more water conservation facilities in hotels and scenic areas should also be built. Third, promoting “green catering” to reduce the waste of food. For example, a better grain conservation standard can be established to encourage tourists to take part in the “Clean Plate Campaign”.Promoting digital tourism to reduce the intensity of resource consumption. First, enriching the tourist experience by introducing virtual reality (VR). Local high-tech industry and scenic spots could work together to promote sophisticated VR/augmented reality (AR) advertising, VR educational study tours, VR tour commentary, VR immersive tours, and VR historical experience museums, etc., to integrate tourism resources, expand the forms of tourism activities, and eventually reduce tourists’ demand for actual resources. Second, establishing and standardizing tourism livestreaming. The government should issue regulations to standardize the behavior of live streamers, the aesthetic taste, marketing, commenting, and interaction of live streaming rooms, so as to improve the cultural value added to sightseeing, and finally create an immersive tourism experience for both “on-the-spot” and “online” tourists.Establishing multiple environmental monitoring systems to urge interested parties to fulfil their responsibilities. First, building an Internet of Things (IoT) system where the local government is the supervisor and the scenic spot the main participant to monitor and collect the information of tourists’ green consumption behavior. Via the system, feedback regarding how the eco-tourism policy operates can be obtained timely and guide enterprises to optimize the product structure accordingly. As a result, government departments, tourism enterprises, and tourists all shoulder their environmental liability. Second, world-famous insurance service providers with great credibility should be introduced to expand and improve the market structure. To be specific, insurance companies could invite colleges and universities, and experts of environmental emergency response, to assess the security risks of insured tourism enterprises, propose detailed suggestions with time limit for rectification, and work with local governments to expand the green-credit-linked environmental liability insurance (ELI) coverage. In doing so, a permanent third-party monitoring system can be established to motivate tourism enterprises to shoulder their environmental liability more actively.

In the face of frequent extreme climatic events, water and air pollution, and other eco-environment issues, it should be regarded as the responsibility of all countries to make tourism development plans with the constraints of resources and environment. It is also an important demonstration of the international community’s efforts to implement the concepts of green development and man-water harmony. Xinjiang is a typical arid and semi-arid region along the Chinese border. It resembles China’s neighboring countries such as Kazakhstan and Tajikistan, in climate, culture, customs, and tourism resources, and they even share river systems, which is the practical foundation to develop the “China-Kazakhstan Cross-border Tourism Cooperation Zone” and “Silk Road Tourist Area”. Therefore, suggestions on tourism development proposed in this paper can not only help Xinjiang’s government to make sustainable development plans, but also serve as a reference for some countries in Central Asia. In addition, this paper constructs a suitability model to determine the range of growth rate of tourist scale under different development scenarios. This method offers a new perspective on the cap of tourist number of tourist destinations, and on how to make sure the growth rate of tourist scale matches the local ecological environment.

## Figures and Tables

**Figure 1 ijerph-20-02224-f001:**
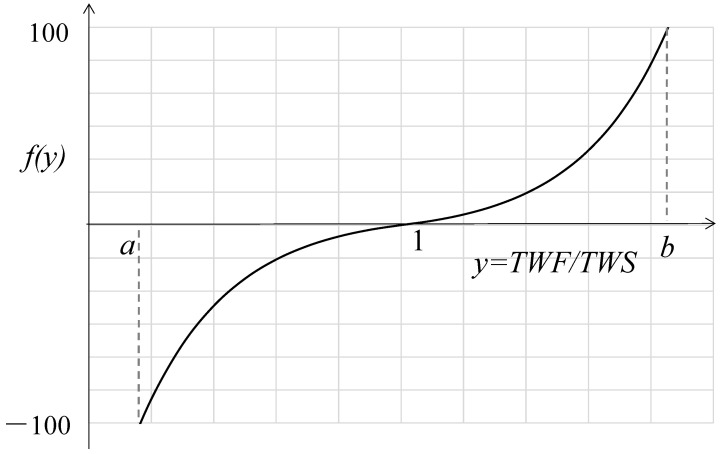
Graph of suitability model.

**Figure 2 ijerph-20-02224-f002:**
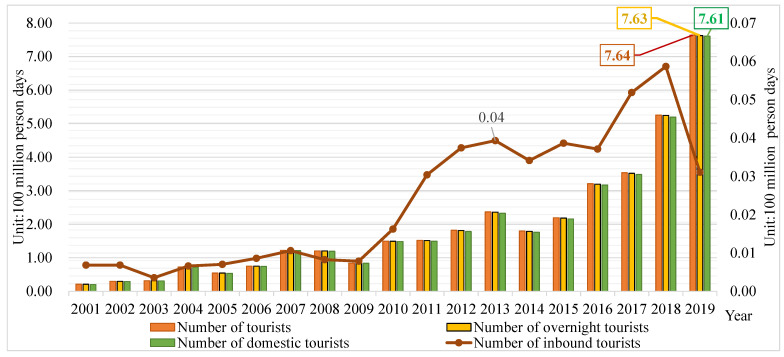
Growth of tourists in Xinjiang during the sample period.

**Figure 3 ijerph-20-02224-f003:**
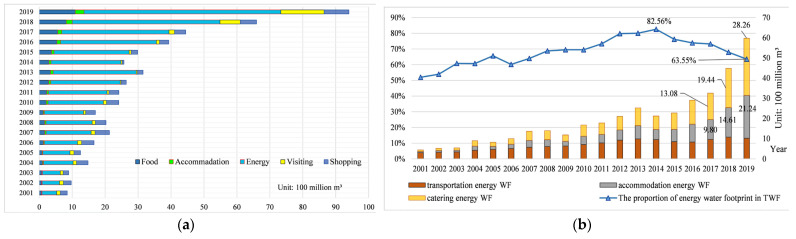
TWF in Xinjiang during the sample period (**a**) and proportion of energy-related water footprint in TWF (**b**).

**Table 1 ijerph-20-02224-t001:** Energy consumption intensity of transportation modes and proportion of tourists in passenger turnover volume [13,46].

Indicator	Railway	Highway	Civil Aviation	Water Way
Energy consumption intensity: *η* (MJ/passenger-kilometer)	1.0	1.8	2.0	1.26
Proportion of tourists in passenger turnover volume (passenger-kilometer):γ(%)	31.6%	13.8%	64.7%	10.6%

**Table 2 ijerph-20-02224-t002:** Calculation methods of water footprint accounts related to food, accommodation, visiting, and shopping [31].

Water Footprint Accounts	Calculation Methods	Description
Food water footprint	$\begin{array}{c} {\mathrm{WF}}_{\mathrm{cater}}=(N_{v}D_{v}+n_{emp}D_{emp})\times \\ (\sum_{j=1}^{n}Food_{j}w_{j}^{food}+0.0015) \end{array}$	$N_{v}D_{v}$ is the total number of tourists. $n_{emp}$ is the number of tourism practitioners. $D_{emp}$ is the average number of working days. j=1,2,3......n is the type of food consumed by tourists every day. $Food_{j}$ is the amount of food *j* consumed by residents (unit: kg/person/day). $w_{j}^{food}$ is the volume of virtual water contained in per unit of food *j*. 0.0015 is the amount of drinking water per person per day (m^3^).
Accommodation water footprint	$\begin{array}{c} WF_{accom}=(q_{v}Q_{v}D_{v}+q_{emp}n_{emp}^{accom}D_{emp}^{accom})\\ +\sum_{k=1}^{n}P_{k}M_{k} \end{array}$	$Q_{v}D_{v}$ is the total number of overnight tourists (the same in Formula (4)). $q_{v}$ is the standard of accommodation water use for each tourist. $q_{emp}$ is the water use standard for hotel staff. $n_{emp}^{accom}$ is the total number of hotel staff. $D_{emp}^{accom}$ is their average working days. $P_{k}$ is the water use standard for producing disposable consumer goods *k*. $M_{k}$ is the number of disposable consumer goods *k* used by the hotel.
Visiting water footprint	$WF_{visit}=g/G\times w_{eco}$	g is tourism consumption. G is local GDP. $w_{eco}$ is water consumed for local ecological environment.
Shopping water footprint	$WF_{shop}=Expend\times w_{indu}$	Expend is tourists’ shopping consumption. $w_{indu}$ is water consumption per 10,000 yuan of industry added value.

**Table 3 ijerph-20-02224-t003:** Suitability category.

Underdevelopment		Suitability	Overload		
A	B	C	D	E	F
Extreme underdevelopment	underdevelopment	No water pressure	Light overload	Moderate overload	Serious overload

**Table 4 ijerph-20-02224-t004:** Assumed tourism scenarios.

	Event	Normal Development	Emergencies Damaging Public Health	Emergencies Damaging Socioeconomic Stability
Scenario				
Scenario Ⅰ		Yes	No	No
Scenario Ⅱ		Yes	Yes	No
Scenario Ⅲ		Yes	No	Yes
Scenario Ⅳ		Yes	Yes	Yes

Note: “Normal development” refers to zero emergencies.

**Table 5 ijerph-20-02224-t005:** Methods to predict indicators related to TWS and TWF [52].

Indicator	Definition	Prediction Method	Detail
*Pop*	The number of residents (100 million person-days)	$\mathrm{Pop}=\frac{k}{1+be^{-at}},\;a\;>0,b\;>0$	Year-end number of residents is in line with the growth curve, so the logistic growth model is applied. *t* represents the forecast year.
*g*	Tourism consumption (100 million yuan)	$g=c_{0}+c_{1}t+c_{2}t^{2}$	The total tourism consumption shows a parabola on the time axis. *t* represents the forecast year.
*G*	GDP (100 million yuan)	$G=(1+\alpha_{G}{)}^{t}\times G_{0}$	$\alpha_{G}$ is the average annual growth rate of GDP in the region during the sample period. $G_{0}$ is the GDP in base year. *t* represents the forecast year.
$n_{emp}$	Tourism practitioners (10,000 person)	$n_{emp}=n_{emp}^{\left(0\right)}+t\Delta n_{emp}$	This paper calculated the average annual increase of the number of tourism practitioners ($\Delta n_{emp}$) according to the progress of local tourism.
$N_{emp}$	Employed population (10,000 person)	$N_{emp}=(1+\alpha_{Nemp}{)}^{t}\times N_{emp}^{\left(0\right)}$	$\alpha_{Nemp}$ is the average annual growth rate of employed population in the region in the sample years. $N_{emp}^{\left(0\right)}$ is the number of employed population in the region in the base year. *t* represents the forecast year.
$WS_{total}$	Annual water supply (100 million m³)	/	It is determined based on development planning in the region.
*T*	Passenger turnover volume (100 million person-kilometres)	$T_{i}={\left(1+\alpha_{i}^{T}\right)}^{n}T_{i}^{\left(0\right)}$	$\alpha_{i}^{T}$ is the average annual growth rate of passenger turnover volume in the sample years. $T_{i}^{\left(0\right)}$ is passenger turnover volume in the base year. i=1,2,3,4 represents railway, highway, civil aviation and water way.
$Food_{j}$	The amount of food *j* consumed by residents (kg/person/day)	$Food_{j}=\sum_{k=k_{0}}^{k_{0}+t-1}Food_{jk_{0}}/t$	Tourists consume the same food as residents. Although the amount and types of food consumed by residents fluctuate on the time axis, they are stable overall, so the moving average method is used to predict tourists’ food consumption.$k_{0}$ represents the starting year. t is the moving average by year. $Food_{j}^{\left(k_{0}\right)}$ is the amount of food *j* consumed by tourists in the starting year.
Expend	Total shopping consumption of tourists (100 million yuan)	$Expend=c_{0}+c_{1}N_{v}D_{v}+c_{2}Inc$	Total tourism consumption is related to the number of tourists and their income level. Therefore, the linear causal model is applied to predict tourism consumption in forecast year.$c_{0}$ is intercept. $c_{1}$ and $c_{2}$ are coefficients. Inc represents tourists’ income level.

**Table 6 ijerph-20-02224-t006:** Predicted values of indicators related to TWS of Xinjiang in 2025.

Indicator	Predicted Value	Indicator	Predicted Value
Pop (100 million person-days)	103.41	$n_{emp}$ (10,000 person)	11.01
g (100 million yuan)	6358.63	$N_{emp}$ (10,000 person)	1639.80
G (100 million yuan)	27,330.83	$WS_{total}$ (100 million m³)	538.45

**Table 7 ijerph-20-02224-t007:** Predicted results of indicators related to TWF of Xinjiang in 2025.

Indicator		Predicted Value	Indicator	Predicted Value
*T* (100 million passenger-kilometers)	Railway	466.34	$\sum_{j=1}^{n}Food_{j}w_{j}^{food}$(m³)	1.50
	Highway	97.40	$w_{p}$(m³/person/time)	0.035
	Civil aviation	372.59	$w_{indu}$(m³)	25.12
	Water way	0	$\sum_{k=1}^{n}P_{k}M_{k}$(100 million m³)	0.0061
$q_{v}$(L/d/person)		350		
$n_{emp}^{accom}$(person)		17,731	$q_{emp}$(L/d person)	90
Inc(yuan)		58,193.09	$n_{food}$(time)	3

Note: Due to the underdeveloped waterway transportation in Xinjiang, the relevant statistical yearbooks do not record the passenger turnover of waterways within the territory, so this paper estimates this indicator in 2025 as 0. Because domestic tourists take a lion’s share, nationwide per capita disposable income of China in 2025 is used to predict tourists’ income level. According to the national plan for building a water-saving society in Xinjiang, water consumption per 10,000 yuan of industrial added value will be reduced by 16% during the 14th Five-Year Plan period. With 2019 as the base year, water consumption per 10,000 yuan of industrial added value in 2025 will be around 29.90 × (1 − 16%) = 25.12 m³. Other related indicator values are predicted based on the tourism progress in Xinjiang.

**Table 8 ijerph-20-02224-t008:** Values of y in different suitability categories.

Variable	Underdevelopment		Suitability	Overload		
	A	B	C	D	E	F
	Extreme underdevelopment	underdevelopment	No water pressure	Light overload	Moderate overload	Serious overload
f(y)	[−100,−50)	[−50,−25)	[–25,25]	(25,35]	(35,60]	(60,100]
$\left[\underline{y},\bar{y}\right]$	[0.1200, 0.4661)	[0.4661, 0.6979)	[0.6979,1.3574]	(1.3574,1.4709]	(1.4709,1.7085]	(1.7085,2.000]

**Table 9 ijerph-20-02224-t009:** Predicted values of total number of tourists, TWS and TWF of Xinjiang in 2025.

Indicator		Lower Limit	Upper Limit
Total number of tourists (100 million person-days)		5.05	13.26
TWS (100 million m³)		103.58	110.84
TWF (100 million m³)	Total tourism water footprint	72.29	150.46
	Food water footprint	7.09	20.26
	Accommodation water footprint	1.78	4.65
	Energy-related water footprint	49.28	102.47
	Visiting water footprint	13.01	13.29
	Shopping water footprint	0.30	9.80

## Data Availability

The datasets used and/or analyzed during the current study are available from the corresponding author on reasonable request.

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
