# Peer review of "Tourism Development under Water-Energy Dual Constraints: A Case Study from Xinjiang Based on Different Emergency Scenarios"

_ijerph, 2023, doi:10.3390/ijerph20032224_

Round 1

Reviewer 1 Report

This paper first constructed the calculation model of tourism water supply (TWS). Second, according to the tourism life cycle theory, the energy-related water footprint account was built and combined with energy and water consumption, to realize water-energy dual constraints. Then, a suitability model between TWS and tourism water footprint (TWF) was established. This study offering new insights for research on TWS , improving TWF account based on a life cycle assessment approach, enriching research methods on the green development of tourism via water-energy dual constraints. And the methods adequately described and the results clearly presented in this paper.

However, the article also has some shortcomings:

Firstly, Figure 2 is not clearly presented.

Secondly, In the article, only railroads, highways and civil aviation are considered in the calculation of transportation energy consumption, ignoring water transport as the main transportation mode.

Thirdly, Tourism development is not only constrained by water and energy consumption, but also by environmental impacts such as solid waste, air pollution and other aspects, so it is necessary to explore the efficiency of tourism development under environmental constraints.

Reviewer 2 Report

Overall, this is an interesting area of research, as the research method of processing is relatively novel. By constructing a calculation model of tourism water supply (TWS); establishing a suitability model between TWS and tourism water footprint (TWF), and predicting the growth rate of tourists in Xinjiang under the “suitability” state between TWS and TWF. In addition, the proposed suggestions are practical. My personal advice for improving the quality of the article is as follows.

(1)China has a variety of tourism resources throughout its territory, however, I personally believe that the authors should go into greater detail about why they have chosen Xinjiang as the research object for this article. Although there is a brief introduction to Xinjiang in the article, by supporting them with numerical data or comparing with other areas will help the study to be done in a way that leaves no doubt.

(2)There are many spelling mistakes in the article. It would be helpful if the authors could double-check this point.

(3)The article is based on the tourism life cycle model, however, the introduction of the theory in the article is not very sufficient, in light of this, it may be beneficial to enhance the readability of this article by introducing this theory. Please add some fresh and relevant literature to enhance the value of your paper.

(4)It is better to add the implications of the article, whether on a practical or theoretical level. 

Reviewer 3 Report

While acknowledging the author's efforts, the paper requires significant revision to warrant publication.

Introduction

The introduction generally reads well except few issues. Given that some research has been done, what is prompting your research? It would be best if you showed a strong justification. Please strengthen your arguments by explicitly pointing out what past research still needs to be done and why it needs to be done. So that you know, doing that will point out your theoretical contribution. Finally, it is difficult to understand the role of the first three sentences in paragraph two. The first paragraph and the subsequent paragraphs make no comparison to the MENA region. I suggest you delete this as there seems to be no connection “In recent years, tourism in arid and semi-arid regions has flourished due to their diverse natural, cultural, historical, and religious resources. For example, tourism in the 34 Middle East and North Africa (MENA) region increased by about 10% in 2018, reaching 35, approximately 6% of the world’s visitor arrivals. In addition, jobs created by tourism in 36 MENA are estimated to rise from about 5.7 million in 2018 to over 7.2 million in 2028 [3]”.

Literature Review

Authors need to present an overview of the literature to highlight the gaps warranting the study. This will help strengthen the arguments in the introduction.

Methods

The methods section looks fine.

Results/Findings

The results/findings section looks fine.

Discussion, theoretical and managerial implications

My biggest issue is the need for a theoretical and practical implications section. This is important for readers to see the substantial contribution to literature. Specifically, your article needs to provide evidence of a substantial contribution to the literature (the study's theoretical and managerial/practical implications need to be more robustly developed in this regard).
